# Anti-Inflammatory and Anti-Hyperuricemic Functions of Two Synthetic Hybrid Drugs with Dual Biological Active Sites

**DOI:** 10.3390/ijms20225635

**Published:** 2019-11-11

**Authors:** Rafa S. Almeer, Sherif F. Hammad, Ola F. Leheta, Ahmed E. Abdel Moneim, Hatem K. Amin

**Affiliations:** 1Department of Zoology, College of Science, King Saud University, Riyadh 11451, Saudi Arabia; 2Department of Pharmaceutical Chemistry, Faculty of Pharmacy, Helwan University, Cairo 11795, Egypt; sherifhammad2010@hotmail.com; 3Clinical Pathology Department, Faculty of Medicine, Suez Canal University, Ismalia 41522, Egypt; 4Department of Zoology and Entomology, Faculty of Science, Helwan University, Cairo 11795, Egypt; 5Department of Biochemistry and Molecular Biology, Faculty of Pharmacy, Helwan University, Cairo 11795, Egypt; hatem.k.amin@gmail.com

**Keywords:** inflammation, xanthine oxidase, paw edema, diclofenac, febuxostat, allopurinol, arthritis

## Abstract

The present study aimed to test the anti-inflammatory and xanthine oxidase inhibitory activities of two synthesized molecules and compare them to routinely prescribed nonsteroidal anti-inflammatory drugs (NSAIDs), such as diclofenac and the serum urate-lowering drug, allopurinol. The anti-inflammatory effects of the designed compounds (A and B) were evaluated in carrageenan (CAR)-induced paw edema in mice. The levels of nitric oxide and myeloperoxidase activity were measured in paw skin using biochemical methods. Additionally, prostaglandin E2 (PGE2), C-reactive protein (CRP), cyclooxygenase-2 (Cox-2), tumor necrosis factor-α (TNFα), interleukin (IL)-1β, IL-2 and IL-10, and monocyte chemoattractant protein-1 (MCP1) were assessed by enzyme-linked immunosorbent assay (ELISA). The expression of inflammation-related genes was confirmed by real-time qPCR. The expression of inducible nitric oxide synthase (iNOS) and nuclear factor-kappa B (NF-κB) were estimated using immunohistochemistry, and xanthine oxidase inhibitory activity was evaluated using an in vitro assay. The results revealed that compounds A and B decreased inflammation, as was observed by a reduction in the elevation of all the tested markers. In addition, the tested compounds markedly decreased paw swelling, mobilization of inflammatory cells, iNOS-, and NF-κB-immunoreactive cells in a mouse model of paw edema. Interestingly, both compounds were potent xanthine oxidase inhibitors as well as Cox inhibitors with higher activity in favor of compound B providing potential dual acting series of anti-hyperuricemic and anti-inflammatory therapeutic agents.

## 1. Introduction

Inflammation is an important adaptive response for defense against harmful stimuli, such as infection, tissue stress, and injury [1]. The inflammation cascade includes elevated permeability of micro-vessels, attachment of circulating cells to the vessels in the vicinity of the injury site, migration of several cell types, and growth of new tissue and blood vessels [2]. However, excessive inflammation is the major cause of the development of many diseases, including cancer, diabetes, non-alcoholic fatty liver disease, inflammatory bowel diseases, and atherosclerosis [3]. Inflammation is tightly regulated by several components that can be categorized as inducers, sensors, mediators, and effectors [1]. Inflammation itself can also be grouped into two categories: acute and chronic inflammations. Acute inflammation is the early reaction of the immune system against foreign microorganisms and tissue damage. It is a rapid and self-limiting process, arbitrated by eicosanoids and vasoactive amines which increase the movement of plasma and leukocytes into the damaged site [4]. On the other hand, in chronic inflammation, a variety of cytokines and growth factors are secreted, resulting in the mobilization of higher order immune cells including leukocytes, lymphocytes, and fibroblasts. The inflammation can lead to persistent malfunction of tissue triggered by these immune cells [5].

Non-steroidal anti-inflammatory drugs (NSAIDs) have been approved for repose from gentle to moderate acute and chronic inflammations. The underlying mechanism of action of NSAIDs in treating inflammation is through the inhibition of the pro-inflammatory enzyme cyclooxygenase (Cox), which is blamed for stimulating prostaglandins biosynthesis. NSAIDs are categorized into a two-group of both conventional nonselective NSAIDs like diclofenac, acting nonspecifically via suppressing both Cox-1 and Cox-2, and selective Cox-2 inhibitors, like celecoxib [6]. However, targeting Cox-1 and Cox-2 to inhibit prostaglandins formation negatively affects the gastroprotective effects exerted by Cox-1, leading to serious and undesired adverse effects in the gastrointestinal tract [7]. Hence, safety and tolerance concerns associated with the use of this class of drugs continue to be an issue for patients and clinicians.

Xanthine oxidase (XOD) is a key enzyme in purine metabolism that facilities the hydroxylation of both hypoxanthine and xanthine in the last two steps of urate biosynthesis in humans causing hyperuricemia [8]. Hyperuricemia refers to abnormally high levels of uric acid caused by overproduction or decreased excretion of uric acid, and is a key inducer of gout, in addition to the excessive formation of superoxide radicals resulting in many pathological conditions including inflammation, hypertension, and atherosclerosis [9]. Although the therapeutic agent allopurinol has been identified as an effective treatment for gout and other disorders in the clinic, undesired adverse effects of these traditional agents, such as bone marrow suppression, allergic reactions, and gastrointestinal and renal toxicities cannot be ignored [10]. The desire to develop a more effective and less toxic XOD suppressors still exists.

The objective of this study was to test the anti-inflammatory and XOD inhibition activities of the two synthesized molecules 2-(2-methoxyphenylamino)-4-methyl-thiazole-5-carboxylic acid (compound A) and 2-(2,4-dimethoxyphenylamino)-4-methyl-thiazole-5-carboxylic acid (compound B). These two selected compounds were previously patented [11] as an ion-channel inhibitor by another research group. After analyzing their structures and testing their homology in comparison to routinely prescribed NSAID, diclofenac, and serum urate-lowering drug, a major homology called febuxostat was found, and with structural homology illustrated in Figure 1. Therefore, these two compounds with their privilege structures were tested as an anti-inflammatory and as serum urate-lowering drugs.

## 2. Results

### 2.1. Acute Toxicity Analysis of Compounds A and B

Acute toxicity study in male mice treated with the different compounds was performed in order to explore adverse effects of administration. A single dose of compounds A and B (10, 20, and 40 mg/kg body weight) administered orally revealed no signs of toxicity in treated male mice. Furthermore, after 14 days of the treatment, no mice died and no significant changes in the body weight or absolute weight of kidney, spleen, stomach, testis, or heart were found in comparison to the control mice with a non-significant decrease in the absolute weight of liver (Table 1). However, the absolute weights of some organs showed significant changes at high dose administration of compound A or B.

### 2.2. Effects of Different Compounds on CAR-Induced Paw Edema

To determine the potential anti-inflammatory effects of compound A and compound B in comparison with the reference anti-inflammatory drug, diclofenac, we used a CAR-induced paw edema model in mice. As shown in Figure 2, compounds A and B showed significant anti-inflammatory activity elicited by the paw volume reduction, and compound B was more active than compound A.

### 2.3. Effects of Different Compounds on CAR-Induced Histopathological Changes

As shown in Figure 3, histopathological examination of paw tissue of CAR-treated group revealed epithelial hyperplasia, inflammatory cell infiltration, and edema. These signs of inflammation were greatly attenuated by compounds A and B. As previously observed, compound B was more active than compound A. Likewise, the anti-inflammatory edema response evoked by compound B was similar to that exerted by diclofenac pre-treatment.

### 2.4. Effects of Different Compounds on CAR-Induced Inflammation

C-reactive protein is widely used as a vascular marker of inflammation. Hence, we determined the levels of CRP in the plasma of mice. CAR injection markedly increased CRP levels compared with the vehicle control group (Figure 4). Mice treated with the two compounds prior to CAR showed a significant decrease in CRP as compared to the CAR-treated mice. The results indicated that compound B had a more potent effect on decreasing the plasma levels of CRP as the reference drug. Thus, the anti-inflammatory properties of the compound B can contribute to the alleviation of edema development.

Injection of CAR on paw significantly elicited an inflammatory reaction in mice (Figure 5), as judged by edema development and leucocyte infiltration that was determined by increasing in the thickness of the paw skin and increased levels of tissue pro-inflammatory cytokines (IL-1β, 2, TNF-α, MCP-1, PGE2, and Cox-2), NO production and MPO activity and decrease in the anti-inflammatory cytokine, IL-10. Interestingly, the tested compounds showed anti-inflammatory activity, which was observed by a significant decrease in the pro-inflammatory cytokines, NO production, and MPO activity and an increase in IL-10 levels. We also observed that compound B reduced paw edema better than a 20 mg/kg dose of diclofenac. These results indicate that the tested compounds possess anti-inflammatory activity, and they can modulate the inflammatory mediators in CAR-induced acute inflammation. Additionally, quantitative real-time PCR (qRT-PCR) analysis confirmed the anti-inflammatory activity of the tested compounds (Figure 6).

Furthermore, we showed an anti-inflammatory effect of Compounds A and B by evaluating iNOS and NF-κB expression in paw tissues. CAR injection caused a robust inflammatory reaction, as was noted by a marked increase in iNOS and NF-κB levels in the paw tissue more than those in the control sections (Figure 3). Compounds A and B pretreated mice showed markedly lower iNOS and NF-κB immunoreactive cells than in the CAR-induced paw edema mice model.

Diclofenac is a commonly used NSAID with a well-established mechanism of action. It inhibits both Cox-1 and Cox-2 enzymes and prostaglandin PGE2 in correlation to the drug concentration in the plasma and induces the anti-inflammatory mediators, IL-10 and transforming growth factor. COX has different isoforms; Cox-1 is the “housekeeping” isoform expressed in most tissue types controlling platelet functions. Cox-1 provides cytoprotection to the gastric mucosa through prostacyclin activation and by regulating renal blood flow. In contrast, Cox-2 expression is elevated in response to tissue damage and proinflammatory mediators, resulting in increased production of pain and inflammation mediators namely, prostaglandin, thromboxane, and leukotriene. Non-selective inhibition of diclofenac for Cox-2 might explain the drug safety for cardiovascular side effects compared with highly selective Cox-2 inhibitors. In human synovial cells, IL-1-induced PGE2 release has been demonstrated by labeling the cells with radioactive arachidonic acid. Diclofenac inhibited radioactivity associated with free arachidonic acid and increased the radioactivity amount related to phosphatidylethanolamine and triglycerides [12,13,14].

The inhibition of the inflammatory mediators and the induction of IL-10 by compounds A and B are similar to the mechanism of action of diclofenac that inhibits inflammation via the inhibition of the Cox-2/Cox-1 enzymes, various inflammatory mediators like IL-1 and TNF-α and prostaglandin PGE2 [12]. Results from our study suggest that strong anti-inflammatory effect of both compounds A and B are comparable to diclofenac due to homology in their active site structures. Histopathological analysis of the sacrificed animals showed that compound B rescued the CAR-induced animal paw inflammatory edema back to the normal animal paw and showed similar anti-inflammatory efficacy as diclofenac, while compound A showed relatively lesser efficacy with residual inflammation and tissue redness.

The two tested compounds, designated compound A and compound B are hybrid molecules with two active sites and are structural analogs to both diclofenac and febuxostat. Both compound A and compound B showed major inhibition of Cox-2, CRP, IL-1β, IL-2, iNOS, NO, MPO, MCP1, TNF-α, and PGE2, and showed a significant anti-inflammatory effect compared with diclofenac (*p* < 0.5) (Figure 7). Treatment with both compound A and compound B increased the levels of the anti-inflammatory interleukin IL-10 compared to the control and diclofenac-treated group (*p* < 0.5).

### 2.5. Xanthine Oxidase Inhibition Properties

The XOD inhibitory effect of compounds A and B was determined and the corresponding data is shown in Figure 8. Compound A showed a 28%, 27%, and 42% XOD inhibition in the activity at 10, 20, and 40 µg/mL, respectively. While compound B showed the highest XOD inhibition of 54%, 42%, and 76% at 10, 20, and 40 µg/mL, respectively. Percent inhibition was determined to be 8%, 14%, and 20% for allopurinol, a clinical XOD inhibitory drug, at 0.5, 1, and 2 µg/mL, respectively.

XOD is an interconvertible form of xanthine oxidoreductase which is abundant in several tissues like gut, kidney, lung, liver, capillary endothelial cells, heart, brain, and plasma and plays a major role in the uric acid production inside the human body. Febuxostat is a thiazole derivative and potent non-purine noncompetitive human XO inhibitor. Its structure-activity relationship with XOD has been investigated and it was revealed that thiazole ring at position 4 has a small hydrophobic substituent and the 3-cyano and 4-isobutoxyphenyl at position 2 of the thiazole ring have hydrophobic and electron withdrawing properties. Both of these two structural features enhance the inhibitory effect of febuxostat on XOD activity through the interaction of thiazoles in XOD binding site compromising of the hydrogen and ionic bonds at Arg880, Thr1010 and Asn768 for the active carboxylate and cyano groups, respectively [15,16].

Although compounds A and B share structure similarity with febuxostat, the serum urate-lowering and anti-gout effects of these tested compounds were assessed against allopurinol for the following reasons; first, allopurinol is a purine analog and inhibits XOD via dose-dependent competition, second, allopurinol primary metabolite is oxypurinol, which is also a xanthine oxidase inhibitor with a half-life of 15 to 18 h. Third, both drug and its active metabolite are excreted in the urine and feces. Hence, evaluating the tested compounds against allopurinol makes it favorable to have a better view of the drug onset, its duration, and the side effects, as allopurinol is a dose-dependent drug and interferes with the purines metabolism.

Both compound A and compound B showed major inhibition of XOD activity indicating a strong serum urate-lowering effect compared to allopurinol. This result could be explained through their structural similarity with febuxostat, the mechanism of action is likely to be similar as described by Semlcervoic et al. [16]. Compound B showed a stronger inhibition of XOD activity than compound A, as its active site is more aligned and similar to febuxostat, allowing it to induce XOD enzyme hydrogen and ionic bonds compromises at Arg880, Thr1010, and Asn768 for the active carboxylate and cyano groups, respectively in the inhibitor compound.

## 3. Materials and Methods

### 3.1. Chemistry Rationale, Synthesis and Analysis

#### 3.1.1. Chemistry Rationale

Figure 1 illustrates the structural homology between the tested compounds and the prototype drugs illustrating the rationale for their design and hence, explains the structure of activity relationships for both of them.

#### 3.1.2. Chemical Synthesis

The following synthesis scheme (Scheme 1) demonstrates the synthesis of the target N-aryl substituted-2-aminothiazole-5-carboxylic acid derivatives 3a and 3b.

#### 3.1.3. General Procedure for the Preparation of Compounds 2A&B

Thiourea (1.00 mmol) was added to a solution of 3-chloropentane-2,4-dione (1.00 mmol) or ethyl 2-chloro-3-oxobutanoate (1.00 mmol) in 20 mL of methanol. The mixture was kept for refluxing overnight. After the solution cooled to room temperature, 5% K_2_CO_3_ was used to neutralize it. Excess solvent was removed under vacuum. The crude product was extracted using ethyl acetate, which was then washed with brine, dried over Na_2_SO_4_, and concentrated in reduced pressure. The final product was obtained by further purification on column chromatography.

#### 3.1.4. General Procedure for the Preparation of Compounds 3A&B

A mixture of corresponding compound 2a or 2b (0.3 mmol), and 1 M NaOH solution (3.0 mL) in tetrahydrofuran/EtOH (1:1) (5.0 mL) was heated at 45 °C for 1.5 h. After completion of the reaction, the solvent was removed under reduced pressure. The residue was dissolved in water (20 mL) and acidified with a 1M HCl solution to pH 3. The solid was collected by filtration, washed with water, and purified by flash column chromatography (0–35% EtOAc in hexanes) to yield the desired product. The structure elucidation of target compounds and their intermediates was confirmed by means of ^1^HNMR and ^13^C-NMR and was in accordance with the patented compounds.

### 3.2. Acute Toxicity Study

Acute toxicity study was carried on male Swiss mice (20–25 g) purchased from VACSERA (Cairo, Egypt). The mice were allowed to acclimate into their new environment for 1-week and maintained in standard clear plastic cages in the animal laboratory. The animals were housed at a temperature between 24 and 26 °C, a normal light: dark cycle, and relative humidity of 50%. They were fed standard rodent chow and water ad libitum and were handled in accordance with the National Institutes of Health (NIH) Guidelines for the Care and Use of Laboratory Animals, 8^th^ edition (NIH Publication No. 85-23, revised 1985) and all the experimental procedures were approved by the Institutional Animal Ethics Committee guidelines for animal care and use at Helwan University (approval no, HU2017/Z/03 in 28 September 2017). Mice were divided into three groups involving the vehicle control and compound A or B including 20 mice each. The mice were fasted for 24 h before administering the dose with access to adequate drinking water. The treated mice were administered the treatment orally at a dose of 0, 10, 20, 40, or 100 mg/kg body weight. Mice from all the groups were observed individually for any signs of toxicity within the first 24 h after the treatment and the vital organs, namely the liver, kidney, heart, spleen, stomach, and testis were collected from two mice of each group. Mortality was assessed in the remaining mice and recorded if any for 14 days after the treatment.

### 3.3. Carrageenan (CAR)-Induced Paw Edema

Male Swiss mice (20–25 g) were divided into four groups: normal; CAR; Diclofenac (Diclo) + CAR (diclofenac, 20 mg/kg) as a standard drug reference; compound A (20 mg/kg) + CAR; and compound B (20 mg/kg) + CAR. The compounds were dissolved in saline and orally administered to the mice for 3 consecutive days before induction of edema. Paw edema was induced by injecting 0.1 mL of 1% *w*/*v* CAR suspended in saline into the sub-plantar tissues of the left hind paw of each mouse. The volume of the paw was determined with a Vernier caliper (LETICA Scientific Instruments, Barcelona, Spain) immediately prior to CAR injection and again at 2, 4, 6, and 8 h after injection. The data were expressed as the variation in the paw volume (mL) and were compared to the right hind paw of the same mouse. At 8 h, animals were euthanized and paw skin samples were collected and divided into two samples. One sample was homogenized immediately to yield 50% (*w*/*v*) homogenate in ice-cold medium containing 50 mM Tris-HCl (pH 7.4) and centrifuged at 500× *g* for 10 min at 4 °C. The supernatant was used for the various biochemical assays, while the second sample was used for the histological or molecular study.

### 3.4. Myeloperoxidase (MPO) Activity Assay

Neutrophil migration to paw was determined based on the MPO activity according to the modified method described by Bradley et al. [17]. After three freeze-thaw cycles of the homogenate and centrifugation at 15,000× *g* for 10 min at 4 °C, MPO activity was determined by mixing 200 μL of the supernatant with 2.8 mL of 50 mM phosphate buffer (pH 6.0) and 1 mL of 1.67 mM *O*-dianisidine hydrochloride containing 0.0005 % (*v*/*v*) H_2_O_2_. The change in the absorbance at 450 nm was recorded, and MPO activity was expressed as U/mg protein.

### 3.5. Measurement of Nitrite/Nitrate Levels

The assay for analyzing the nitrite/nitrate levels in paw supernatant was performed according to the method of Green et al. [18] by adding the Griess reagent (a mixture of naphthylene diamine dihydrochloride (0.1%) and sulfanilamide (1% in 5% H_3_PO_4_)) for 10 min in dark at 30 °C, and the absorbance of the bright reddish-purple azo dye was measured at 540 nm.

### 3.6. Cytokine and Mediator Analyses

The levels of prostaglandin E2 (PGE2), C-reactive protein (CRP), Cox-2, IL-1β, 2 and 10, TNF-α and MCP1 were assessed by ELISA using anti-mouse PGE2 (Cat. No, CSB-PA040059), Cox-2 (Cat. No, CSB-E12910m), MCP1 (Cat. No, CSB-E07430m), TNF-α (Cat. No, CSB-E04741m), IL-1β (Cat. No, CSB-E04621m), 2 (Cat. No, CSB-E04627m) or 10 (Cat. No, CSB-E04594m) antibody (CUSABIO Life Sciences, Wuhan, China) according to the manufacturer’s instructions.

### 3.7. In Vitro Analysis of XOD Inhibitory Activity 

Xanthine oxidase inhibitory activity of the compounds was determined in vitro spectrophotometrically by measuring uric acid formation at 295 nm at 25 °C. The method was based on the procedure described by Kostić et al. [19] with some modifications. The assay mixture contained 100 mM sodium pyrophosphate buffer (pH 8.3), 0.3 mM Na_2_EDTA, 0.064 mM xanthine, 15 U/L XOD (Sigma, X4376, St. Louis, MO, USA), and the test compound. The XOD inhibition by various compounds was assayed by the reduction of the uric acid formation. The enzyme needed pre-incubation for 15 min at 30 °C with the test compound, and the reaction was started by addition of substrate and xanthine. Allopurinol was used as a positive control. All the reactions were performed in triplicates.

The equation reported by Sweeney et al. [20] was used to evaluate the degree of XOD inhibitory activity, wherein α is the activity of XOD without test compound and β is the activity of XOD with the test compound. 

% XO inhibition = (1 − β/α) × 100(1)

### 3.8. Histopathological and Immunohistochemical Analysis

The hind paws skin was fixed in 4% neutral buffered formalin and then after 24 h, samples were embedded in paraffin, sectioned into 4–5 μm thick sections, and stained with hematoxylin and eosin (H&E) to analyze the general histopathological changes. Immunoreactivities of inducible nitric oxide synthase (iNOS) and NF-kB profiles were investigated using purified primary antibodies with avidin-biotin-peroxidase (ABC) and peroxidase substrate (Pierce™ Peroxidase IHC Detection Kit, Thermo Fisher Scientific, Waltham, CA, USA). Briefly, the endogenous peroxidase activity was blocked by 0.3% H_2_O_2_ for 30 min in a humidity room after heating (95–100 °C) the sample. The sections were then incubated with primary antibody overnight at 4 °C in a humidified room followed by incubation with biotinylated rabbit anti-mouse secondary antibody (Dako system kit, Santa Clara, CA, USA) and avidin-biotin complex (ABC) reagents for 1 h at 30 °C in a humidified room. Finally, the specimens were counterstained with hematoxylin, dehydrated, and mounted using Aquatex fluid (Merck KGaA, Darmstadt, Germany).

### 3.9. Quantitative Real-Time PCR

Total RNA was extracted, and first strand cDNA was synthesized according to the manufacturer’s instructions. The expression of *Il1b*, *Il2*, *Il10*, *Ptgs2* (Cox-2), *Ccl2* (MCP-1), *Tnf*, and *Nos2 (iNOS)* was determined using real-time quantitative reverse transcription polymerase chain reaction (qRT-PCR) technique using an Applied Biosystems 7500 Instrument (Thermo Fisher Scientific, Ottawa, Ontario, Canada). The thermal conditions for qRT-PCR were initial denaturation at 94 °C for 2 min, followed by 40 cycles of 94 °C for 30 s and 60 °C for 30 s, and a final extension at 72 °C for 10 min. After PCR amplification, the Δ*C*t from experiments repeated at least three times was determined by subtracting the *C*t value of the reference gene, glyceraldehyde 3-phosphate dehydrogenase (*Gapdh*) from that of each sample (*C*t). The validated primers sequences of different genes are provided in Table 2.

### 3.10. Statistical Analyses

All data are expressed as the mean ± standard deviation (SD). One-way analysis of variance (ANOVA) was performed to compare the control and treatment groups using the statistical package SPSS, version 17.0 (IBM, Chicago, IL, USA). A *p*-value < 0.05 was considered statistically significant.

## 4. Study Limitations

The results obtained in the present study demonstrate that the synthesized compounds are promising for potential future applications in developing new anti-inflammatory and serum urate-lowering agents, but further molecular, pharmacologic, and toxicologic experiments are needed to confirm the safety and efficacy of these compounds, to gain more insight into the precise underlying mechanism of action, and to determine the optimal dose, route, and formulation for administering of the designed compounds.

## 5. Conclusions

Both compound A and compound B as synthesized hybrid molecules with dual active sites showed a pronounced anti-inflammatory and serum urate-lowering effects in comparison to diclofenac and allopurinol, respectively.

## Figures and Tables

**Figure 1 ijms-20-05635-f001:**
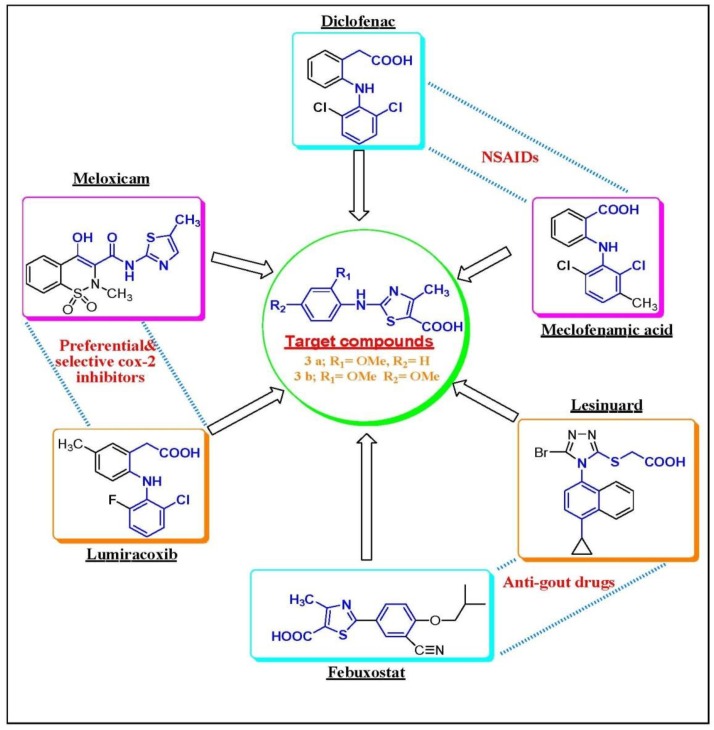
Rational drug design through molecular hybridization approach.

**Figure 2 ijms-20-05635-f002:**
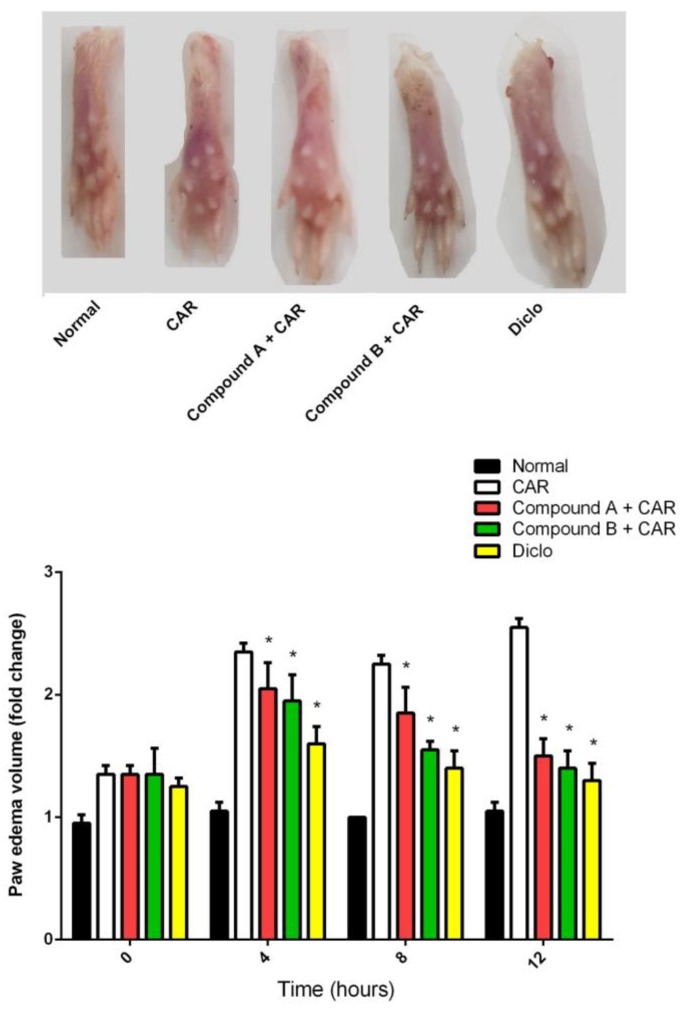
Effect of compounds A, B or diclofenac (Diclo) on paw edema volume in carrageenan (CAR)-induced paw edema in mice. Data are represented as mean ± SD (*n* = 7); *p* < 0.05 indicates statistical significance; * significant change versus the CAR group.

**Figure 3 ijms-20-05635-f003:**
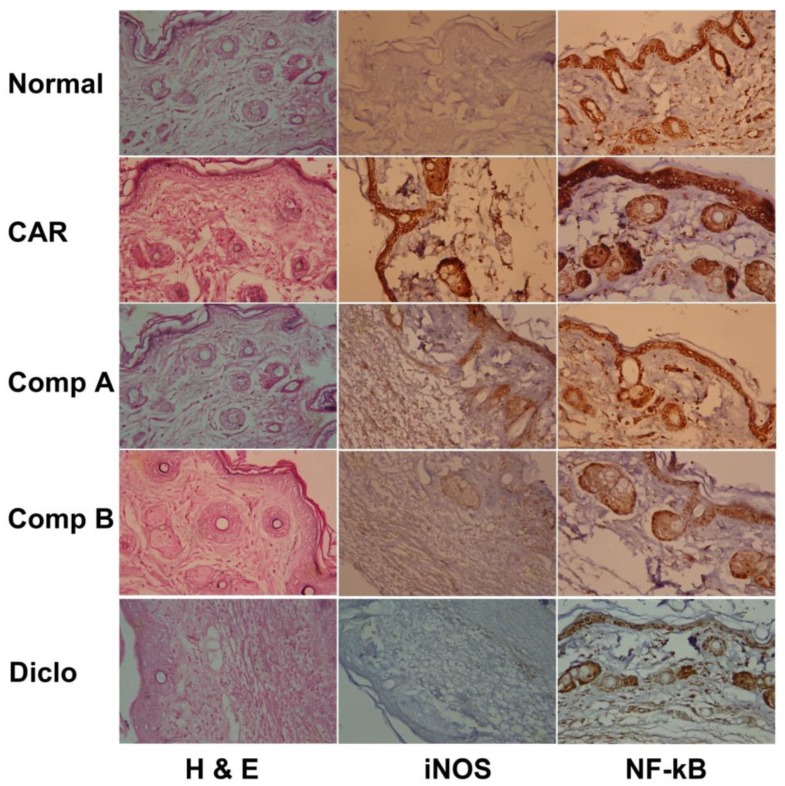
Effect of compounds A, B, or diclofenac (Diclo) on paw skin histology and iNOS and NF-κB expression detected by immunohistochemistry in carrageenan (CAR)-induced paw edema in mice (Original magnification 400×).

**Figure 4 ijms-20-05635-f004:**
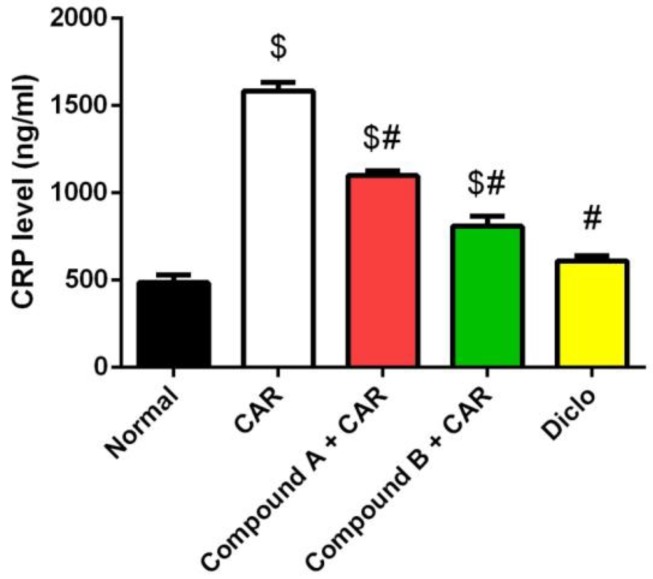
Effect of compounds A, B, or diclofenac (Diclo) on C-reactive protein level (CRP) in carrageenan (CAR)-induced paw edema in mice. Data are represented as mean ± SD (*n* = 7); *p* < 0.05 indicates statistical significance; $, significant change versus normal mice; #, significant change versus the CAR group.

**Figure 5 ijms-20-05635-f005:**
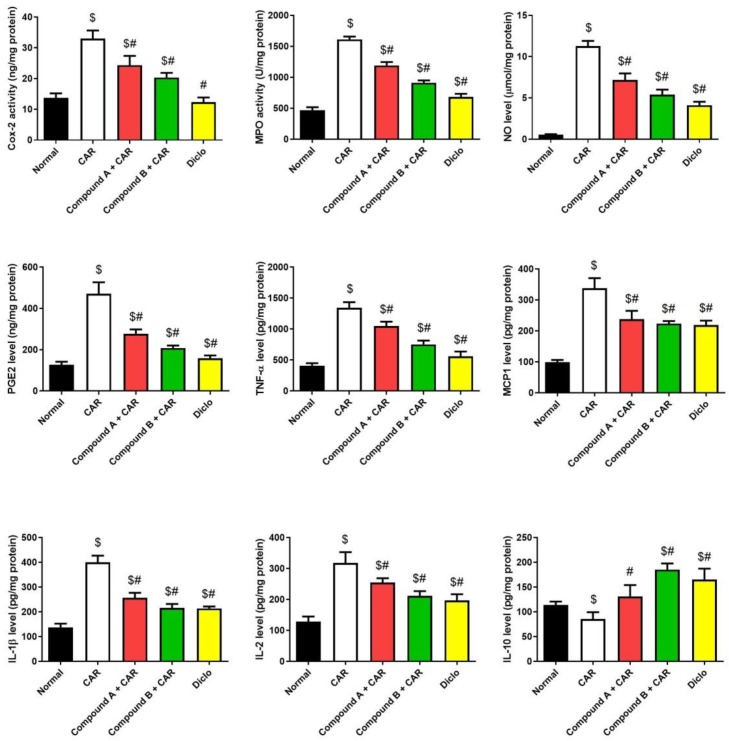
Effect of compounds A, B or diclofenac (Diclo) on pro-inflammatory markers in carrageenan (CAR)-induced paw edema in mice. Data are represented as mean ± SD (*n* = 7); *p* < 0.05 indicate statistical significance; $, significant change versus normal mice; #, significant change versus the CAR group.

**Figure 6 ijms-20-05635-f006:**
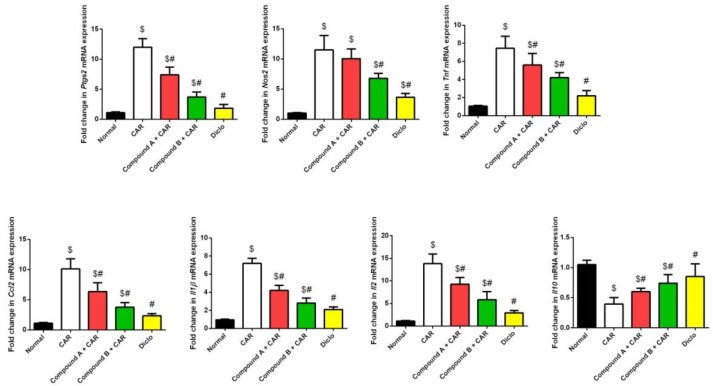
Effect of compounds A, B or diclofenac (Diclo) on pro-inflammatory markers mRNA expression in carrageenan (CAR)-induced paw edema in mice. mRNA results (mean ± SD of three independent assays) were normalized to the *Gapdh* mRNA levels and are shown as fold induction relative to the mRNA levels in the control.; *p* < 0.05 indicates statistical significance; $, significant change versus normal mice; #, significant change versus the CAR group.

**Figure 7 ijms-20-05635-f007:**
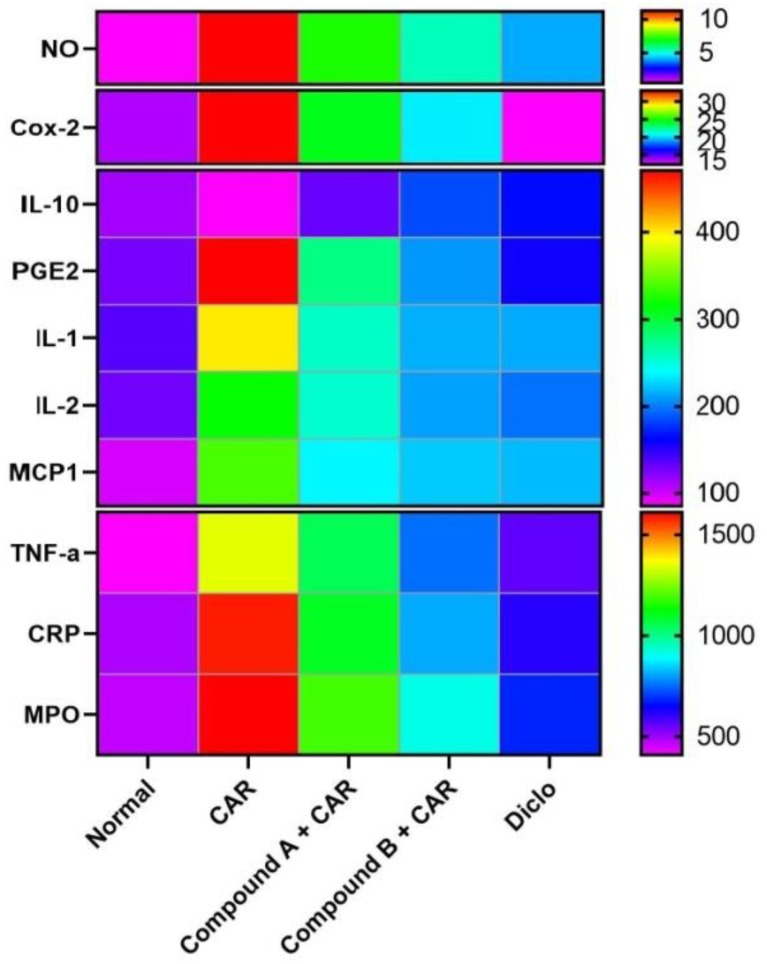
Heatmap showing the effect of each tested compound on the tested inflammatory mediators with diclofenac.

**Figure 8 ijms-20-05635-f008:**
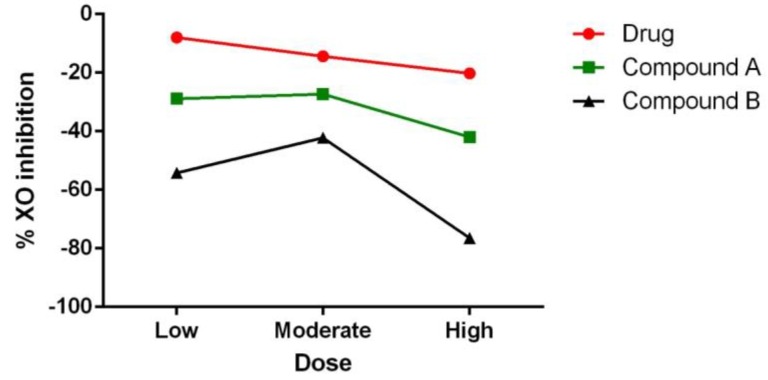
Inhibitory effects of different concentrations of Compounds A, B, or allopurinol on the activity of xanthine oxidase.

**Scheme 1 ijms-20-05635-sch001:**
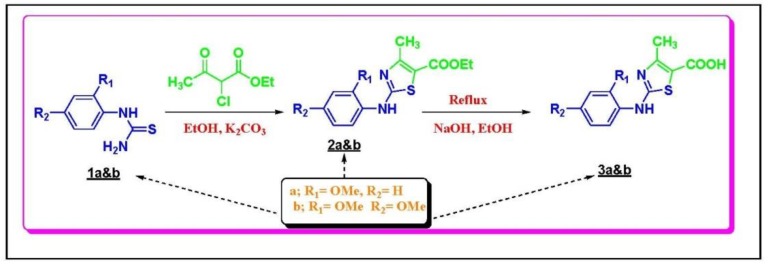
Synthesis of the target N-Arylsubstituted-2-aminothiazole-5-carboxylic acid derivatives 2a and 2b.

**Table 1 ijms-20-05635-t001:** Absolute organ weights of male mice treated with Compounds A and B after 14 days.

Compound Dose (mg/kg bwt)		Liver (g)	Kidney (g)	Heart (mg)	Spleen (mg)	Stomach (g)	Testis (mg)
Vehicle		1.83 ± 0.21	0.73 ± 0.14	208.61 ± 27.36	146.34 ± 16.28	0.82 ± 0.19	192.71 ± 20.54
Compound A	10	1.86 ± 0.18	0.73 ± 0.15	212.54 ± 31.05	151.28 ± 17.16	0.82 ± 0.20	201.34 ± 19.76
	20	1.85 ± 0.23	0.74 ± 0.13	201.36 ± 25.48	155.68 ± 14.91	0.84 ± 0.18	206.15 ± 18.66
	40	1.87 ± 0.17	0.75 ± 0.17	218.64 ± 33.15	140.19 ± 20.38	0.85 ± 0.21	201.51 ± 20.69
	100	1.90 ± 0.25 *	0.78 ± 0.16	201.43 ± 26.27	163.17 ± 18.06 *	0.81 ± 0.21	226.17 ± 22.35 *
Compound B	10	1.84 ± 0.16	0.74 ± 0.17	209.14 ± 21.60	138.78 ± 15.67	0.83 ± 0.20	204.32 ± 16.87
	20	1.87 ± 0.22	0.76 ± 0.19	215.32 ± 31.47	148.14 ± 20.13	0.84 ± 0.22	214.81 ± 20.39
	40	1.86 ± 0.20	0.76 ± 0.15	219.44 ± 24.61	151.32 ± 17.42	0.84 ± 0.21	218.36 ± 21.65
	100	1.95 ± 0.27 *	0.79 ± 0.18	227.64 ± 30.21 *	174.18 ± 21.36 *	0.87 ± 0.28 *	236.14 ± 17.69 *

Values are expressed as the mean ± standard deviation. Statistically different from the vehicle control group; * *p* < 0.05.

**Table 2 ijms-20-05635-t002:** Primer sequences of genes analyzed in real time PCR.

Name	Accession Number	Sense (5′–3′)	Antisense (5′–3′)
***Gapdh***	NM_001289726.1	CCCATCACCATCTTCCAGGAGC	CCAGTGAGCTTCCCGTTCAGC
***Ccl2***	NM_011333.3	GCAGCAGGTGTCCCAAAGAA	ATTTACGGGTCAACTTCACATTCAA
***Il2***	NM_008366.3	TGAGTCAGCAACTGTGGTGG	GCCCTTGGGGCTTACAAAAAG
***Il10***	NM_010548.2	ATAACTGCACCCACTTCCCA	GGGCATCACTTCTACCAGGT
***Il1b***	NM_008361.4	CCTTCCAGGATGAGGACATGA	TGAGTCACAGAGGATGGGCTC
***Nos2***	NM_001313922.1	CGAAACGCTTCACTTCCAA	TGAGCCTATATTGCTGTGGCT
***Ptgs2***	NM_011198.4	CAGACAACATAAACTGCGCCTT	GATACACCTCTCCACCAATGACC
***Tnf***	NM_001278601.1	ACCCTCACACTCACAAACCA	ACCCTGAGCCATAATCCCCT

The abbreviations of the genes; *Gapdh*: Glyceraldehyde-3-phosphate dehydrogenase; *Ptgs2*: Prostaglandin-endoperoxide synthase 2 (Cox-II); *Nos2*: nitric oxide synthase 2, inducible; *Il1b*: Interleukin 1 beta; *Tnf*: Tumor necrosis factor; *Ccl2*: Chemokine (C-C motif) ligand 2 (MCP-1).

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
