# Peer review of "Anti-Inflammatory and Anti-Hyperuricemic Functions of Two Synthetic Hybrid Drugs with Dual Biological Active Sites"

_ijms, 2019, doi:10.3390/ijms20225635_

Round 1

Reviewer 1 Report

Manuscript Rafa S. et. al. - thematically fits the IJMS and is interesting, but a larger group of compounds should be analyzed so that better analysis and conclusions can be drawn towards the design of new and better drug.

Author Response

Thanks for your positive comments, we have already synthesized a larger group of related compounds and they were assessed in the same direction of biological activities emphasizing the dual anti inflammatory and anti gout potentials. What we were aiming for is to proof the concept and the two selected compounds were representative and selected based on structural relevance to each other.

Reviewer 2 Report

I’ve read with attention the paper of Almeer et al. that is potentially of interest. The methodology applied is overall correct, the results are reliable and adequately discussed. I’ve only some minor comments:

- The discussion section should include some lines on the study limitation and study perspectives

- Anti-hyperuricemic drug should be "serum urate-lowering drugs"

- The paper requires an attentive revision of the whole text and figures because of a number of typos

Author Response

- The discussion section should include some lines on the study limitation and study perspectives

Response: Thanks for your comments and we added study limitations as you suggested.

- Anti-hyperuricemic drug should be "serum urate-lowering drugs"

Response: We followed the comment.

- The paper requires an attentive revision of the whole text and figures because of a number of typos.

Response: We carefully revised the work to avoid any typo errors. Additionally, the work was already edited by a professional service to abolish its English.